# Examining the Intended and Unintended Impacts of Raising a Minimum Legal Drinking Age on Primary and Secondary Societal Harm and Violence from a Contextual Policy Perspective: A Scoping Review

**DOI:** 10.3390/ijerph18041999

**Published:** 2021-02-19

**Authors:** Ruud T. J. Roodbeen, Rachel I. Dijkstra, Karen Schelleman-Offermans, Roland Friele, Dike van de Mheen

**Affiliations:** 1Tranzo Scientific Center for Care and Wellbeing, Tilburg University, P.O. Box 90153, 5000 LE Tilburg, The Netherlands; R.Friele@nivel.nl (R.F.); H.vdMheen@tilburguniversity.edu (D.v.d.M.); 2Netherlands Institute for Health Services Research (NIVEL), P.O. Box 1568, 3500 BN Utrecht, The Netherlands; 3Tilburg Law School, Tilburg University, P.O. Box 90151, 5000 LE Tilburg, The Netherlands; R.I.Dijkstra@tilburguniversity.edu; 4Netherlands Institute for the Study of Crime and Law Enforcement, P.O. Box 71304, 1008 BH Amsterdam, The Netherlands; 5Faculty of Psychology & Neuroscience, Department of Work & Social Psychology, Maastricht University, P.O. Box 616, 6200 MD Maastricht, The Netherlands; karen.offermans@maastrichtuniversity.nl

**Keywords:** minimum legal drinking age, alcohol policy, policy impact, societal harm, societal violence, public health, underage youth, environmental research, scoping review

## Abstract

Raising a minimum legal drinking age (MLDA) has generated interest and debate in research and politics, but opposition persists. Up to now, the presentation of impacts focussed on effectiveness (i.e., intended impact); to our knowledge, no literature syntheses focussed on both intended and unintended impacts. A systematic scoping review was conducted in which a search strategy was developed iteratively and literature was obtained from experts in alcohol research and scientific and grey databases. Ninety-one studies were extracted and analysed using formative thematic content analysis. Intended impacts were reported in 119 units of information from the studies (68% positive), forming four paths: implementation, primary and (two) on secondary societal harm and violence. Unintended developments were reported in 43 units of information (30% positive), forming five themes. Only eight studies reported on implementation. Furthermore, a division between primary and secondary paths and the use of a bridging variable (drinking patterns in analyses or methodology) was discovered. These results provide an insight into how well legislation works and can be used to discover or implement new means of curbing underage drinking and alcohol-related violence and harm. They also offer valuable starting points for future research and underline the importance of considering unintended developments.

## 1. Introduction

Raising the minimum legal drinking age (MLDA) has generated much interest and debate in research and politics over the past decades due to the conduct of numerous investigations, the multiplicity of impacts that have been found and the moral sensitivity surrounding the debate. The general intention of raising an MLDA (e.g., from 18 to 21 years of age) is to further decrease the availability of alcohol for underage adolescents which, in turn, is expected to reduce alcohol use and its associated harm to adolescents and their environment [1,2,3,4,5]. Despite confirmed evidence of the effectiveness of raising the MLDA found in multiple studies and literature reviews (e.g., [5,6]), opposition to a higher MLDA persists, especially in the United States [6,7,8,9,10].

An extensive body of literature exists in reviews on the impact of increased MLDA [5,6,9,11,12,13,14]. These reviews have presented the effects of an increase in the MLDA on reduced drinking patterns and other alcohol-related harm and violence. Reviews have also indicated that an increase in the MLDA protects underage drinkers from short-term negative outcomes (e.g., being involved in an alcohol-related traffic crash [13]) as well as long-term negative outcomes (e.g., alcohol and other drug dependence in adulthood [6]). Furthermore, it is argued that although the magnitude of effects may appear small, these effects apply to the entire population of youth and therefore result in very large societal benefits [5]. Up to now, the approach to presenting the impact of an increase in the MLDA has mainly focussed on the effectiveness of the changed policy (i.e., the intended impact). To our knowledge, no literature synthesis of previous studies has focussed on the intended as well as the unintended impacts of a raised MLDA. Moreover, the use of a comprehensive theoretical model to present this information and give more insight into the contextual aspect is lacking.

Responsive and realism evaluation (theories used for the general evaluation of legislation) indicate the importance of a contextual perspective when changes in legislation occur [15,16,17,18]. Both theories consider the sometimes complex, capricious and unintended relationship between legislation, on the one hand, and reality, on the other, when changes in legislation occur. They show that all forms of knowledge, action and process (and not only the most general or intended effects) should be investigated and used to understand the true impact of legislation [15,16,17,18]. Therefore, unintended impacts and the processes or developments that in reality occur after an increase in the MLDA should be considered as well. Any form of impact should be taken into consideration during the evaluation and justification of changes in legislation, whether this is a positive change resulting in attainable benefits or a negative change showing an opposite impact. In this study, intended impact of raised MLDA is defined as a direct decrease in the availability of alcohol for underage adolescents and the sequential reduction of alcohol use and associated harms to adolescents and their environment. Unintended impact refers to all additional processes, developments or occurrences in reality caused by raised MLDA.

A good starting point to further substantiate the perspective needed in this study is the conceptual model for raising an MLDA introduced by Lanza-Kaduce and Richards [19]. According to these authors, the most important value of their model (see Figure 1) is the ability to present the unintended as well as the intended impacts that a change in policy can have.

The authors started their model with a “policy”, which was the increase in 1985 of the MLDA in Florida from 19 to 21 years. They assumed that the primary policy objective of this increase (the last step in the model) was to reduce the frequency of youthful drink-driving behaviour and/or the level of impairment of youthful drinking-drivers (for the target population of 19- to 20-year-olds) [19]. Two directions were then possible. Starting with “unintended impact”, they examined whether the increase in the MLDA led to derivative lawbreaking, whether the change engendered a sense of injustice and whether the social context of drinking was altered. “Implementation” described the process for 19- and 20-year-olds in Florida born before 1966 that were “grandfathered in”, meaning that this specific group of underage youth were allowed to drink during the year that the higher MLDA was introduced. Furthermore, according to the authors, if increasing an MLDA is to affect drink-driving behaviour, it must deter drinking and/or restrict the opportunity to drink so that individuals drink less. Therefore, they constructed a “bridging variable” that measured drinking patterns of respondents and preceded the primary policy objective of the increase (the last step in the model) [19].

Although this model is acceptable for the presentation of intended and unintended impacts of a change in policies, the individual aspects were unidimensional and were only focused on the Florida setting. Yet policy objectives or implementation processes from other settings could be relevant as well. In addition to drink-driving behaviour, alcohol-related traffic crashes could be an example of an important type of additional societal impact. Furthermore, “implementation” should not be limited to “grandfathering in” effects but should include all processes or developments after an increase in the MLDA that aid implementation and change impacts in reality.

Our aim is to present a broad spectrum of intended as well as unintended units of information derived from the literature relative to the impact of a raise in the MLDA on primary and secondary societal harm and violence using the conceptual model proposed by Lanza-Kaduce and Richards [19]. The outcome of this study is a novel and empirically-based overview of all impacts after an MLDA was raised. By presenting impacts this way, we are able to provide insight into how well legislation works and use this to discover new means of curbing underage drinking and alcohol-related violence and harm. Also, these insights offer valuable starting points for future research and underline the importance of considering unintended developments. Furthermore, this overview could further help calibrate the ways in which professionals advocate, develop, implement, evaluate and legitimise changes in policy aimed at curbing alcohol availability [20]. A scoping review will be conducted synthesising what is known from relevantliterature.

## 2. Methods

### 2.1. Aim, Purpose and Design

Scoping reviews are an ideal tool to determine the scope or coverage of a body of literature on a given topic, providing an overview (broad or detailed) of the literature’s focus [21]. The purpose of our study matches two generally accepted purposes for conducting scoping reviews [21,22,23,24] and follows the protocol proposed by Arksey and O’Malley [24] and using the additional advancements of this protocol by Levac, Colquhoun and O’Brien [23].

### 2.2. Search Strategy

#### 2.2.1. Phase 1

The scientific databases of the Web of Science, Sociological abstracts, PubMed, PsycINFO and Embase were searched using different strategies for an initial scope of the scientific literature (May and June of 2019). No timespan was selected for the search. Search strategies were developed iteratively per database. When applicable, the initial search started with categorisations developed by the database (e.g., using MeSH terms in PubMed, or using Emtree in Embase). In addition, free text terms were applied iteratively, searching the literature more broadly. When applicable and in line with the research question of our study, Boolean operators (AND, OR, NOT), parenthesis, truncation (e.g., raise*, increase*) or additional proximity searching was used in order to facilitate the search strategy. The final search strategies are included in Appendix A. 

#### 2.2.2. Phase 2

The finalised strategies used in phase 1 were adapted for searching grey literature databases (e.g., theses, reports, policy documents) using OAIster, GLIN, Opengrey and Google Scholar (July of 2019). The final search strategies are included in Appendix A.

#### 2.2.3. Phase 3

National and international experts in the field of alcohol research were invited to indicate relevant literature (scientific or grey) in English or Dutch languages. In August of 2019, an invitation was sent to the members of the Kettil Bruun Society (https://www.kettilbruun.org/; accessed on 11 October 2019 [25]), who are specialised in social, epidemiological and cross-cultural research on alcohol policy and use (the invitation is included in Appendix A).

### 2.3. Selection Process

All literature was uploaded into Endnote software and, after merging data, duplicates were removed [26,27]. The literature was screened and selected by one author (R.R) based on title (e.g., excluding titles containing the word “tobacco” or titles describing medical laboratory studies). The same author then conducted an initial screening in which all literature was reviewed more thoroughly based on title and abstract. Independently, another author (R.D) reviewed a randomly selected 10% of all studies (77 of 740 studies) using initial criteria. For three of the 77 studies (4%), disagreements had to be resolved through discussion between the authors. There was no need to change the initial criteria. Thereafter, the second author (R.D) screened the other 90% of studies (663 studies) and final disagreements were resolved for 23 out of 663 studies (3%) in a meeting between both authors (R.R, R.D). During all steps in the process described above, co-authors were repeatedly consulted to resolve differences (K.S.O., R.F., D.v.d.M.). The inclusion criteria had been finalised in consultation with all authors.

During the selection of studies, literature reviews were set aside. After completing the selection, a full-text evaluation of reviews was conducted by one author (RR). Reviews were deemed relevant if they (partially) focused on the impact of a raised MLDA. The studies included in the reviews regarding the impact of a raised MLDA (and meeting the criteria listed below) were added to those included in this study [5,6,9,11,12,13,14]. References from search results obtained from databases and studies generated from key reviews were hand-searched by one author for additional relevant studies and added to the included studies when relevant (RR).

#### General Criteria for Exclusion

The selection of studies and the development of criteria followed an iterative process. Post-hoc additions of criteria are a central element to the scoping review process, as it is unlikely that researchers will be able to identify parameters for exclusion at the beginning [23]. Since the general aim of this study is a comprehensive overview of intended and unintended impacts, a selection criteria solely based on the methodological quality of studies (e.g., as is performed by Wagenaar and Toomey [5]) is, in our view, incomplete. Adhering solely to these criteria could unwillingly exclude studies with methodological disadvantages that may have relevant contextual information. Therefore, inclusion criteria for quality selection used in realist reviews were implemented in the development of criteria in this study. These criteria assess studies for their *relevance* (i.e., whether the study addresses the theory under test) and for their *rigour* (i.e., whether an interference has sufficient weight to make a methodologically credible contribution) [28,29].

In general, studies were excluded when they did not target or were not related to increases in an MLDA. For example, studies only focused on outlet density (e.g., [30]) or only focused on reducing BAC-levels (e.g., [31]) were excluded. Furthermore, in general, studies were excluded when they were performed in areas smaller than a state or province (e.g., [32]), or only targeted a specific subgroup in society (e.g., college students, military [33,34]). A full description of definitive exclusion criteria is included in Appendix A.

### 2.4. Data Extraction

One author (R.R) performed a full text assessment of the included literature. Details were extracted and recorded regarding: (1) the aim of the study; (2) the design used (target population); (3) the policy measures or interventions under investigation; (4) the measured policy effects (type of impact and statistical significance or relevance); (5) the reflection on policy and other occurrences by authors; and (6) the data source used (when applicable). Studies were also checked and extracted by the co-authors when one of them (K.S.O., R.F., D.v.d.M.) had doubt regarding the precision of the extraction, leading to an approximate 10% of double extracted studies by co-authors. Differences in extracted data were adjusted or added to the extraction.

Regarding the fourth category (measured policy effects), in quantitative studies, impact was recorded if an effect was found to be statistically significant in the measured association at hand. In qualitative studies, a realist perspective was used using relevance and rigour criteria as a recording impact in this category [28,29]. Furthermore, for every type of impact extracted, the existence and/or type of “bridging variable” was recorded (alcohol use of underage adolescents). Criteria for deciding whether studies had used these bridging (intervening) variables were: (1) that studies controlled for/mediated for this behaviour in their analysis; (2) that studies used data selection tactics to account for intervening behaviour; or (3) that studies used actual BAC-levels (e.g., [35,36]). Studies that used police-reported indicators for drinking or that selected single vehicle night-time accidents as proxies for alcohol use when investigating alcohol-related motor vehicle accidents among young drivers (e.g., [37,38,39]) were not recorded as bridging variables due to the unreliability of these proxies for drinking [40,41,42].

### 2.5. Formative Thematic Content Analysis

For all included studies, the fourth and fifth categories of the extracted data were analysed using a formative (deductive as well as inductive) thematic content analysis [43]. The pieces of material that represented intended or unintended impact (e.g., the significant or relevant policy effect) or a reflection on policy and other occurrences related to the policy effect, were defined as “units of information”.

All the extracted data within the fourth and fifth categories were read carefully by one author (R.R), and initial codings were applied to the units of information, using the model proposed by Lanza-Kaduce and Richards [19] as a starting point. During coding, multiple forms of societal impact were identified (e.g., not only drink-driving behaviour, but also, for example, alcohol-related traffic crashes) and not all studies were consistent in using “bridging variables” that, according to Lanza-Kaduce and Richards [19], precedes the policy objective of the increase. These findings have initiated the need for extending the model by adding multiple paths that allow the presentation of all forms of impact found in the included literature. This resulted in an overview of four paths that present the multiple forms of impact (including the division between primary and secondary societal impact) and one path presenting themes of unintended impact. During all steps described above creating this overview, as well as the finalization of naming, positioning and describing themes and paths, co-authors of this study were consulted in meetings (RD, KSO, RF, DvdM). In these meetings, the findings were checked, adjusted and supported following an iterative pathway through this type of thematic analysis. 

## 3. Results

### 3.1. Flowchart

A total of 1025 studies were identified from the scientific databases (see Figure 2). The grey search resulted in the identification of 799 studies. Forty-six studies provided by experts were additionally included. Merging studies from databases and experts after removing duplicates resulted in a remainder of 1164 studies. After the first title-screening, 740 studies were selected. An independent review of the titles and abstracts of these 740 studies by two researchers resulted in a remaining 70 studies for full-text assessment. From the 15 literature reviews that had been set aside, 7 key reviews were selected, from which 53 relevant studies were obtained, resulting in 123 relevant studies for full-text evaluation. Seventy-seven studies were deemed relevant and 46 were excluded because inclusion criteria did not apply. After hand-searching the references of 77 included studies, 104 studies were selected for extraction. Although we consulted the main authors of studies that were missing, as well as a professional librarian, and requested studies through interlibrary networks, 13 studies were not available in full text, resulting in 91 studies for extraction.

### 3.2. Significant and Relevant Impacts found in Studies after an Increase in MLDA

Building on the conceptual model described by Lanza-Kaduce and Richards [19], Figure 3 presents the significant and relevant unintended and intended impacts found in the included studies. The number of units of information found in the 91 included studies that reported intended impacts was 119, forming four paths. In 81 units of information, positive impacts were reported (68.1%), marked with a plus sign in Figure 3. Some studies were used in more than one path (and sometimes more than once within a path), since in some cases these studies investigated or reported on multiple types of impacts (e.g., [44,45]).

Information on significant/relevant positive impacts was reported on: ○1st path: implementation (eight units of information, four times positive impact; 50%);○2nd path: primary societal impact (on drinking/purchasing patterns; 37 units of information, 24 times positive impact; 64.9%);○3rd path: secondary societal harm and violence without bridging variable (48 units of information, 35 times positive impact; 72.9%);○4th path: secondary societal harm and violence with bridging variable (26 units of information, 18 times positive impact; 69.2%).

Forty-three units of information that reported unintended impacts were found; in 13 units of information, positive impacts were reported (30.2%). Five themes were reported.

The most important addition to the current model was the division between primary societal impacts (drinking, possession or purchasing patterns) and secondary societal harm and violence (i.e., sequential impacts on primary drinking behaviour, e.g., alcohol-related traffic accidents) and the division between secondary societal harm and violence measured with or without the preceding bridging variable (i.e., without consideration of drinking patterns in analyses or methodology). All paths were described, presenting impacts and relevant study characteristics. Extracted data per study is included in Appendix A, in order of appearance in this results section.

#### 3.2.1. First Path: Implementation

Eight included studies reported information (eight units of information) on processes or developments that occurred after an increase in an MLDA that helped implementation [44,45,46,47,48,49,50,51]. Five studies were conducted in the United States, two in the Netherlands, one in Canada. Six studies used survey research, one study conducted statistical analysis on existing databases and one study used mystery-shopping to gather the information on implementation. One study found that an increase in the strength of “False ID Use Laws” (as part of the increase in the MLDA) was associated with a significant 7.3% reduction of younger-than-age-21 drivers involved in fatal crashes who had a positive BAC [46]. Another study—measuring compliance by alcohol sellers using 15-year-old mystery shoppers—found that after the increase of the MLDA from 16 to 18 years, mean alcohol compliance rates significantly increased when 15-year-olds attempted to purchase alcohol [47]. In two separate studies in which high school principals and prevention workers in addiction-care were interviewed, results showed that both groups perceived no changes in underage drinking, alcohol-related problems or illicit drug use since the increase in the MLDA [45,50]. In two studies, interviews with enforcement officers indicated that the intensity of enforcement was low, sporadic and varied, caused by the lack of personnel, competing priorities and minimal support for the increased MLDA [44,51]. Lastly, after the increase in the MLDA, the perceived parental approval of alcohol use for underage respondents decreased and appeared to correspond to the drinking status (i.e., illegal or legal) instead of the age of the respondent [48,49].

#### 3.2.2. Second Path: Primary Societal Impact

Thirty-five studies were found which reported information (37 units of information) on the impacts of an increase in an MLDA on primary societal impacts (i.e., drinking, possession and purchasing patterns of alcohol) [44,45,49,50,51,52,53,54,55,56,57,58,59,60,61,62,63,64,65,66,67,68,69,70,71,72,73,74,75,76,77,78,79,80,81]. Thirty-seven units of information were found, since 2 of the 35 studies reported information on multiple types of primary societal impacts, finding significant/relevant impacts in some cases and finding no impact in others. In total, 29 studies were conducted in the United States, three in Canada, one in Belgium, one in the Netherlands and one that focused on several European countries. Twenty-seven studies used surveys to gather information on primary societal impact, seven studies conducted statistical analysis on existing databases and one study used a qualitative survey.

Of these 35 studies, 29 (82.9%) found a significant and relevant impact from an increase in the MLDA on primary societal impacts [44,45,49,51,52,53,54,55,56,58,59,61,62,63,64,65,67,68,70,71,72,73,74,76,77,78,79,80,81]. More specifically, of these 29 studies, 14 found a negative (protective) impact on various short-term output measures of drinking patterns (i.e., alcohol consumption for underage youth) after the increase in the MLDA [45,53,54,55,56,58,61,62,64,65,71,74,76,79]. Short-term impact was reported on:○past month alcohol use [45,57,66,78,80,81];○the number of drinking occasions in the past week, month, year or lifetime (lifetime includes age of onset) [56,59,71,79];○binge-drinking, heavy/frequent episodic drinking (i.e., drinking five or more drinks in a row in the last two weeks or per occasion) [53,56,57,60,66,69,74,78,80].

Furthermore, in 4 of the 14 studies mentioned above, a joint impact on drinking patterns was found with increase in the MLDA and the real prices of beer [71,76], Zero Tolerance laws [53] and excise taxes, mass media campaigns, grassroots movements and variations in the implementation of policies over time [54]. Furthermore, out of the 29 studies, 2 studies reported a negative (protective) impact on long-term decreases in drinking patterns after the MLDA was raised [49,77]. Long-term impact was reported on:○past month alcohol use ten years after the enactment of the 21 MLDA [49];○frequent heavy weekend drinking ten years after the enactment of the 21 MLDA [49];○proportion, use and (days of) binge drinking in the past months for adults who were 18- to 20-years-old when the MLDA-environment changed [77].

Furthermore, 5 of the 29 studies found that the increase in the MLDA was associated with significant changes in alcohol purchasing behaviour by teenagers; declines were reported in the frequency of teenagers’ alcohol purchases in on- and off-premise outlets or public places [44,51,62,68,75]. Increases were reported in the frequency of teenagers’ obtaining alcohol at parties and having others purchase alcohol for them [44,51,62]. Additionally, 8 of the 29 studies found that raising an MLDA had a significant decrease on aggregate or per capita alcohol sales [54,58,63,65,67,73,76,77].

Compared to the abovementioned studies which presented impacts, 8 of the 35 studies found no significant or relevant impacts on drinking patterns associated with an increase in the MLDA [44,50,51,57,66,69,75] (one additional study found no significant impact on illicit drug use associated with an increase in the MLDA [60]):○estimated average drinks on a daily basis [51];○the number of drinking (and illicit drug use) occasions in the past week, month or year [44,55,61,70];○the number of binge-drinking days or occasions in the past month (i.e., drinking five or more drinks per occasion) [52,61,70];○proportion of weekly, lifetime and binge drinkers [69];○perceived drinking of underage youth by prevention workers in addiction care [50].

Reasons for not finding any impacts may include limitations in methods, measuring instruments or robustness in analytic models such that they are not able to capture the complexities of multiple alcohol-control policies or relevant risk factors in adolescents that determine relevant output measures [61,64,70]. Other studies point to the lack of impacts regarding raised MLDA due to the influences by other (alcohol) policy changes or communal developments (e.g., Zero Tolerance-laws, age-of-majority laws for birth control access or improved car safety measures) [44,51,52,55]. Lastly, one study indicated that the lack of impacts might be due to the ease with which underage youth are still able to obtain alcohol when going out due to non-compliance by alcohol sellers or secondary supply [50].

#### 3.2.3. Third Path: Secondary Societal Harm and Violence without the Bridging Variable

Forty-eight studies were found which reported information (48 units of information) on the impacts of an increase in the MLDA on secondary societal harm and violence (i.e., sequential impact on, for instance, drink-driving behaviour or traffic accidents) without the bridging variable (i.e., without a consideration of drinking patterns in analyses or methodology) [37,38,39,40,42,44,45,51,62,67,71,74,77,79,80,82,83,84,85,86,87,88,89,90,91,92,93,94,95,96,97,98,99,100,101,102,103,104,105,106,107,108,109,110,111,112,113,114]. Forty-seven studies were conducted in the United States and one in Canada. For gathering information on secondary societal harm and violence, all 48 studies conducted statistical analysis on existing databases (e.g., data on fatal traffic accidents from the Fatality Analysis Reporting System (FARS)). Of the 48 studies, 35 (72.9%) found significant and relevant impacts from an increase in the MLDA, reporting that raising an MLDA was associated with a significant or relevant decrease [37,38,40,71,74,77,79,80,82,83,84,85,87,88,89,90,92,93,94,96,97,98,99,100,101,102,103,104,105,106,107,108,109,110,111]. Thirteen out of 48 studies (27.1%) found no significant or relevant impact associated with an increase in the MLDA [39,42,44,45,51,62,67,86,91,95,112,113,114]. Thirty-nine studies investigated secondary societal harm and violence on traffic accidents [37,38,39,40,42,44,45,62,67,71,74,77,80,82,83,84,85,86,87,88,89,90,91,92,93,94,95,96,97,98,99,100,101,102,103,104,105,106,107] and 9 studies investigated a variety of indicators [51,79,108,109,110,111,112,113,114].

Of the 39 studies investigating traffic accidents, 30 studies (76.9%) found significant and relevant impacts from an increase in the MLDA, reporting that raising the MLDA was associated with a significant or relevant decrease in traffic accidents [37,38,40,71,74,77,80,82,83,84,85,87,88,89,90,92,93,94,96,97,98,99,100,101,102,103,104,105,106,107]. Different types of traffic accidents were investigated; 19 studies reported impacts on traffic fatalities [71,74,77,82,84,85,87,92,93,94,96,97,98,99,101,102,103,106,107], while 11 studies analysed traffic accidents from a broader perspective (i.e., looking at all alcohol-related crashes including property damage instead of just fatal crashes [37,38,40,80,83,88,89,90,100,105,106]. A joint impact with other alcohol control policies was found in some studies, for example, an increase in the MLDA accompanied by the impact of beer taxes, seatbelt laws or dram-shop laws [66,87,98,104,107]. Furthermore, although most studies focused on underage youth (sometimes using older age groups as a control, e.g., [85]), some studies focused on the effects on the entire age-population, investigating multiple age categories [40,66,82,93,101,104]. Furthermore, three studies only investigated or only found impacts from the increase of an MLDA on traffic accidents for males and not for females [72,105,106]. Lastly, two studies reported impacts on long-term decreases in traffic accidents after an increase in the MLDA [77,89], finding a significant 16% lower rate of involvement in traffic accidents over a 6-year period [89] and a significantly lower degree of night-time traffic fatalities for male adults who had not been able to legally drink when the MLDA-environment changed while they were adolescents [77].

Nine out of 40 studies (22.5%) found no significant or relevant impacts on traffic accidents associated with an increase in the MLDA [39,42,44,45,57,62,86,91,95]. Reasons for not finding an impact related to variables in the analyses included implementing drinking experience in analyses [39,91], using proportional (instead of numerical) measures in analyses when investigating traffic accidents [42] and controlling for a corresponding shift in increased crashes from a lower to a higher age group [86]. According to the authors of these studies, raising an MLDA seems to primarily postpone fatalities [91] and additional attention should be directed to the role of driving experience instead of alcohol [39,86,91]. Furthermore, landmark improvements in the accident avoidance and crash protection of cars and advances in medical technology [62], the inability to measure sensitive changes [45] and the resistance to change of 18- and 19-year-olds after an increase in an MLDA (who had previously been allowed to drink) were recognised as reasons for not finding impacts [44]. The gradual approach of increasing an MLDA in New York (increasing their MLDA from 18 to 19 in 1982 and from 19 to 21 in 1985, in contrast to a state such as Michigan, which abruptly increased its MLDA from 18 to 21 in 1978) was recognised as a reason for not finding impacts [95]. Lastly, reasons for not finding an impact after an increase in an MLDA from 18 to 20 in Massachusetts (in comparison to the existing MLDA of 18 in New York) was believed to be due to the stability of the drinking age in New York over several decades [67].

The impact on secondary societal harm and violence without a consideration of drinking patterns (bridging variable) on a varied arrangement of subjects other than traffic accidents was investigated in nine studies [51,79,108,109,110,111,112,113,114]. Five out of nine studies (55.6%) found significant and relevant impacts from an increase in the MLDA, reporting that the raise was associated with a significant and relevant reduction in youth suicide [108], sexually transmitted disease rates [109], categories of violent death (e.g., suicide, homicide) [110], teen childbearing rates [111] and the prevalence of low birth weight, Apgar scores and premature births [79]. Four out of nine studies (44.4%) found no impact from an increase in the MLDA concerning birth outcomes [112], accidental injury other than traffic accidents [51], homicide or suicide [51,113] and drowning [114]. Reasons for not finding an impact were related to omitted factors and secular trends unrelated to changes in an MLDA that affected outcomes [112], the dependence of an increase in the MLDA on other aspects in the cultural environment [113], the possibility of a negligible role of alcohol in aquatic settings [114] and the incapability of establishing a relation between alcohol and nontraffic accidents, homicides and suicide in analyses [51].

#### 3.2.4. Fourth Path: Secondary Societal Harm and Violence with the Bridging Variable

Twenty-six studies were found which reported information (26 units of information) on the impacts of an increase in the MLDA on secondary societal harm and violence with the bridging variable (i.e., with a consideration of drinking patterns in methodology or analyses) [35,36,44,45,46,49,55,62,64,67,69,80,82,100,115,116,117,118,119,120,121,122,123,124,125,126]. Twenty-four studies were conducted in the United States, one in Canada and one in Belgium. Twenty studies conducted a statistical analysis on existing databases and six studies used surveys to gather information.

The bridging variable included in the analyses is based on:○self-reported and (to be) convicted drink-driving behaviour measured on an individual level [44,45,49,57,62,75,100,119,120,123,125];○criteria for alcohol abuse or dependence (and other illegal substances) on an individual level [117];○the incidence of hospital-based health service use (diagnostic codes) linked to alcohol use on an individual level [69];○mean Blood Alcohol Concentration measures (BAC) mostly on an individual level [35,82,116,122];○controlling and/or mediating for drink-driving behaviour mostly on a population level [36,46,59,80,115,118,121,124,126].

Out of 26 studies, 18 (69.2%) found significant and relevant impacts from the increase in the MLDA [36,44,46,49,64,67,80,82,115,116,117,118,119,120,121,122,125,126]. Of these 18 studies, seven found a significant or relevant decrease in traffic accidents [36,46,59,82,121,122,126] and four of these discovered a joint impact of an MLDA with other alcohol control policies (e.g., mandatory seat belt laws, Zero Tolerance laws, 55-mph maximum speed limit) in significantly reducing traffic fatality rates [36,121,122,126]. Of the 18 studies, 11 found impacts on a varied arrangement of subjects [44,49,67,80,115,116,117,118,119,120,125]. Seven studies found a significant reduction in drink-driving behaviour (e.g., driving after drinking or drunk-driving convictions) by young adults due to increased MLDA policies [44,49,62,75,119,120,125]. One of the studies found that changes in the MLDA were significantly related to prenatal drinking, and, in this context, an MLDA of 18 (instead of a higher MLDA) was associated with a significantly higher prevalence of adverse outcomes among births of young mothers (e.g., low birth weight, premature births) [115]. Lastly, three studies found impacts from the increased MLDA on high school dropout [118], the occurrence of vandalism and disorderly conduct [116] and the likelihood to meet criteria for alcohol or illicit drug disorders in adulthood [117].

Out of 26 studies, 8 (30.8%) found no significant or relevant secondary societal harm and violence associated with an increase in the MLDA [35,45,57,64,80,100,123,124]. Four of these studies [45,57,80,124] did find significant or relevant primary societal impacts, with a decrease in drinking patterns in the target population associated with an increase in the MLDA. Furthermore, three out of these eight studies involved the investigation of traffic accidents [35,57,124]. Some reasons for not finding an impact were the additional landmark improvements in the accident avoidance and crash protection of cars, advances in medical technology [62] and a postponement in accidents by youth in the 18–20 category until they reached the age of 21 or older (i.e., postponing traffic deaths, not avoiding them) [124]. Three studies measured arrest data of DUI-offenders [45,100,123]. Two of them identified reasons for not finding an impact on the inability of measuring sensitive changes in the larger context which influences drink-driving behaviour, precluding simple before–after comparisons [45,100]. The third study argued that no impact was found due to teenagers drinking more in cars as opposed to drinking in a tavern or bar [123]. Lastly, one reason for not finding an impact after the increase of an MLDA on the incidence of hospital-based health service use linked to alcohol was that the MLDA-policy was not directed at or did not influence youth needing hospital care for injuries and neuropsychiatric conditions linked to alcohol [69].

### 3.3. Unintended Developments and Implications

Thirty-seven studies reported information (43 units of information) on unintended developments and implications [35,38,40,44,45,47,49,50,51,56,57,58,61,62,65,67,68,69,71,73,82,83,84,87,92,93,95,97,98,102,104,110,115,116,117,118,123]; 33 were conducted in the United States, 2 in the Netherlands, 1 in Canada and 1 in Belgium.

#### 3.3.1. Comprehensive Impact on Adolescents and Commodities

The impact of raising the MLDA seems to be widespread and substantive, going beyond the target population affected and commodities focused on by the increase [35,47,53,68,81,92,93,110,115,117,118]. First of all, studies show that raising an MLDA not only impacts the target population (e.g., youths directly affected by the increase in the MLDA) but has a wide-ranging impact on younger and older age groups as well [35,47,53,68,92,93,110,115]. These studies show that raising an MLDA appears to initiate an additional protective mechanism for young adolescents under an MLDA (i.e., 18-year-olds). For instance, after a raise in the MLDA to 21, 18-year-olds show significantly lower long-term prevalence rates of alcohol consumption and purchasing rates compared to 19- and 20-year-olds [73]. In another study, raising an MLDA showed an additional protective effect on young adult populations characterised by high environmental and genetic risks for drinking from all aspects of their lives (e.g., problematic alcohol use by their parents) [117,118]. Furthermore, results from another study indicated that an increased MLDA not only helps to create a climate of societal disapproval for alcohol use but also for all types of drug use [56].

#### 3.3.2. Limited Impact on Excessive Elements, Subgroups and in General

Problematic drinkers and heavy crimes as excessive elements in society seem unaffected from impacts generated by an increase in the MLDA. The findings from one study showed that the drink-driving behaviour of heavy drinkers is less affected by an increase in the MLDA than the drink-driving behaviour of moderate drinkers [38]. In another study, which measured the incidence of hospital-based health service use for adolescent injuries and neuropsychiatric conditions linked to alcohol (i.e., problematic drinkers), time trends were not found [69]. Furthermore, one study found that among multiple crime types, the effect of the increase in the MLDA increased with the decreasing severity of the crime [116]. Two studies pointed out that an increased MLDA does not lead to a reduction in binge drinking among college students [57,65]. Associated with limited impact in general, a small increase amount (e.g., increasing an MLDA from 18 to 19) could minimalise impact in contrast to a more abrupt increase from 18 to 21 [45,95].

#### 3.3.3. Substitution of Behaviour (Change in Patterns)

The impact of raising an MLDA seems to have substitution effects on commodities, drinking locations and sources of supply used. First of all, one study found that an increased MLDA had the unintended impact of increasing the prevalence of marijuana consumption [56]. Furthermore, three studies found a decrease in underage youth obtaining alcohol in on- and off-premise locations after raising an MLDA [44,51,62] and a doubled social or secondary supply of alcohol. Furthermore, another study found an increase in the number of alcohol-related arrests among people under 20 after an increase in the MLDA [123]. The authors argued that this could be due to teenagers not honouring the new law by drinking more in cars as they were unable to legally drink in a tavern or bar. This substitution effect is also mentioned in a Dutch study, indicating that 16- and 17-year-olds engaged in more drinking out-of-sight on private property after the MLDA was raised [50]. In another study, authors argued that youth cannot use alcohol at home in the presence of their parents after an increase in the MLDA and therefore might use alternative drinking locations such as at their friends’ houses where parents may be absent [49]. Furthermore, “border hopping” was identified in multiple studies [40,56,66,83,84,102]. Border hopping occurs when individuals travel to neighbouring states or provinces with a lower MLDA than their own. More specifically, for counties more than 25 miles from a lower MLDA border, raising the drinking age within a state showed a negative and statistically significant effect on the likelihood that an underage driver is involved in a fatal accident [84]. Farther from such a border, results showed that restrictions caused by an increase in the MLDA were effective in reducing accident fatalities [84].

#### 3.3.4. Interdependence of Policy

Changes in federal excise taxes or stricter DUI penalties could influence (or could be influenced by) an increase in the MLDA. One study found that the price sensitivity for youth alcohol use fell after the change to a uniform MLDA of 21 in the United States [61]. Similarly, another study argued that after an increase in an MLDA, the tax-instrument had less of an impact on youth drinking and fatalities [87]. From a more general viewpoint, the authors formulated that communities with relatively strong existing policies might expect a smaller impact, while communities with weak current regulations might expect larger benefits from identical policy initiatives [87]. On the other hand, some authors have proposed that alcohol policies may interact advantageously. Multiple studies reported that increasing an MLDA appeared to proportionately reduce more accidents when taxes were high compared with when taxes were low, suggesting that alcohol policies may work synergistically [63,66,97]. One of these studies proposed that the additional benefit of combining an increase in the excise tax with an increase in the MLDA is to affect a wider age group (i.e., affecting youths between 15–17 and 21–24 as well) [71]. As a potential downside, the authors argued that a tax increase may greatly stimulate the demand for illegally produced beer [71]. Another study argued that another downside of an increase in the MLDA and beer tax is the exposure of people who drink responsibly to punitive action (i.e., increased taxes) [104]. Lastly, one study suggested that an increase in the MLDA and the imposition of stricter DUI penalties were both responsible for a decline in fatal crashes among underage drivers [82].

#### 3.3.5. Policy Endogeneity and Reverse Causality

The impact of an increase in the MLDA seems to be influenced by policy endogeneity and reverse causality. One study, which investigated the compliance of alcohol vendors, noticed a rise in compliance was already present in the years preceding the introduction of the increased MLDA [47]. According to the authors, this could signify a process in which the general acceptability of juvenile drinking had already started to lower before the increased MLDA was introduced [47]. In addition, the impacts of the inducement of the federally mandated transition to a uniform MLDA of 21 in 1986 in the United States could reveal the occurrence of policy endogeneity. Results of one study suggested that the greatest reduction in traffic fatality rates of youth between 18 and 20 years old occurred in states that adopted the policy on their own (without federally mandated inducements) [62]. These “early adopters” may have enacted a higher MLDA in response to grassroot concerns about drink-driving behaviour or may have devoted significant resources to enforcement [62]. This underlines the importance of local support for a successful implementation of federally mandated law. Furthermore, two studies argued that ignoring the process of endogeneity results in an underestimation of the impact [98,104]. Furthermore, while reflecting on mixed results from the analyses of the impacts of an increase in the MLDA, the authors point to reverse causality as a possible cause, arguing that countries with a higher proportion of lifetime drunkenness are bound to institute a higher MLDA [69].

## 4. Discussion

Prior work has presented evidence for the effects of MLDA increases on reduced alcohol consumption and other alcohol-related harm and violence, protecting underage drinkers from short- and long-term negative outcomes [5,6,9,11,12,13,14]. Up to now, the impacts of an increase in the MLDA has mainly been presented by focussing on intended impacts. To our knowledge, there is no literature synthesis that focusses on intended as well as unintended impacts by using a comprehensive theoretical model to present this information and give a better insight into contextual aspects. For this reason, the aim of this study is to synthesise exactly that, presenting a novel and empirically based overview of all impacts after an MLDA has been increased.

Building on the conceptual framework described by Lanza-Kaduce and Richards [19], we identified four paths of intended impacts, in which 119 units of information were found in the 91 included studies (positive impact was reported in 68% of the units of information). Our results show a gap in current literature: only eight studies reported information on the implementation process of an increase in an MLDA. This is unexpected given the importance of implementation in allowing changes in legislation to function in society [18]. Furthermore, the importance of implementation is underlined in key reviews on the subject of existing MLDA [5,6,13], by implications voiced by authors in the studies included in this review (e.g., [44,47,49,76] and in prior research showing that increased enforcement and compliance improves the effectiveness of existing MLDAs [44,127,128,129,130]. In addition, implementation involves more than instruments deployed by governments; it involves all the processes or developments within all levels of society that occur in reality after the increase in an MLDA. This could additionally involve, for instance, the emergence of self-regulation among alcohol sellers [131,132], the handling of social or secondary supplies of alcohol by alcohol sellers [133], or the development of a risk-oriented approach of deploying enforcement and prevention efforts more effectively [134]. Future research should try to address this gap in knowledge and focus on the implementation process surrounding an increase in an MLDA.

A division between primary and secondary societal impacts was made in order to organise the impacts found in the included studies. Furthermore, a division between studies on secondary societal harm and violence was made with and without the bridging variable. This addition follows the reasoning of Lanza-Kaduce and Richards [19] for adding a bridging variable to their model. In our review, we found that 48 studies reported information on secondary societal harm and violence without a bridging variable, that is, without taking drinking patterns into account. Limitations on the availability of data have mostly forced the researchers of these included studies to omit drinking patterns or use proxies in their analysis and methodology. In some cases, studies used potentially unreliable proxies as bridging variables for drinking patterns (e.g., [40,41,42]) and were therefore included in the path without a bridging variable. It is thus important that research evaluating an increase in an MLDA (possibly commissioned by governments) starts before the new policy is introduced so that future research relevant to alcohol outcome measures can be included when assessing the processes of secondary societal harm and violence. In this study, we identified five bridging-variable categories representing drinking patterns in analyses or methodology when assessing secondary societal harm and violence. Future research should try to investigate the possible influence of different categories of bridging variables on impacts. For instance, it is not unlikely that the methodological validity of alcohol-related traffic-fatality measures will improve when, as a bridging variable, alcohol use is mediated for (e.g., [126]) instead of controlled for (e.g., [36]) in analyses.

We identified five themes of unintended developments using 43 units of informationfrom the 91 studies (30% of the units of informationreported positive impacts). Using these themes, we provided an additional approximate expectancy of occurrences when an MLDA is raised. For instance, we may predict that when an MLDA is raised, not only is the target population affected, but also older or younger individuals. It is also likely that when an MLDA is raised, problematic drinkers as a vulnerable group in society remain unaffected. Furthermore, in the context of an increase in the MLDA, the substitution of other sources of supply or the potential of border hopping to neighbouring provinces, states or countries with a lower MLDA is likely. From a more general perspective, we have learned that policy endogeneity has the potential to affect the impacts of an increased MLDA in a positive way and that interdependence with an existing alcohol-control policy is possible. 

Our comprehensive literature search, theory-driven approach and rigorous methods can be seen as the strengths of our paper. However, some methodological considerations need to be discussed. First of all, in this review, in quantitative papers, we have recorded an impact if an effect was found to be statistically significant in the measured association at hand. Although we have also included relevant results from qualitative studies from a realist perspective [28,29], we cannot rule out the reflection and reproduction of publication bias. However, because our units of information not only consisted of significant or relevant policy impact, but also on additional observations by the authors of included studies on other policy and occurrences related to impact, this broad perspective might have reduced publication bias. Secondly, although we have searched for grey literature using the appropriate search engines, insights on raised MLDA in local results using native language are perhaps omitted and therefore absent in our review. Part of this is prevented by inviting national and international experts in the field of alcohol research to indicate relevant literature. Thirdly, in this review, the majority of results (and conclusions) are based on data from the United States, since most research on MLDA policy is conducted in this region. This could have created an incline towards results from a homogeneous context. We believe that the context in which a MLDA is raised matters, and therefore, more evidence on raised MLDA policy from other contexts is needed (e.g., the European context). Lastly, we have focused on raised MLDA policy to investigate the impact when changes in alcohol policy occur. Yet, MLDA are lowered as well. Future research should additionally address impact of lowered MLDA to further substantiate the findings in this review.

The information gained from this scoping review not only more accurately supports the assessment of impacts and offers valuable starting points for future research, but also provides an insight into how well legislation works. This underlines the importance of considering unintended processes surrounding legislation instead of solely focussing on intended effectiveness. Furthermore, these insights can be used to discover or implement new means of curbing underage drinking and alcohol-related violence and harm and could additionally aid legislators to further calibrate the ways in which they advocate, develop, implement, evaluate and legitimise changes in policy aimed at curbing alcohol availability [20]. For instance, if an area with neighbouring regions aspires to raise the MLDA, legislators should foresee border hopping [84] and prioritise enforcement activities to these regions. Moreover, after an increase in an MLDA, legislators can expect modifications of impacts to be generated by their tax-instrument [61].

## 5. Conclusions

This study has provided a novel and empirically-based overview of intended as well as unintended impacts after a raise in the MLDA. This overview offers the possibility of considering any type or form of impact during the evaluation and justification of changes in legislation. Whether this involves impacts following the paths or unintended impacts, can be considered and used to estimate how legislation will function in society. As a consequence, positive impacts can be emphasised and negative impacts can be toned down to ultimately protect adolescents and their environment from alcohol-related harm and violence.

## Figures and Tables

**Figure 1 ijerph-18-01999-f001:**
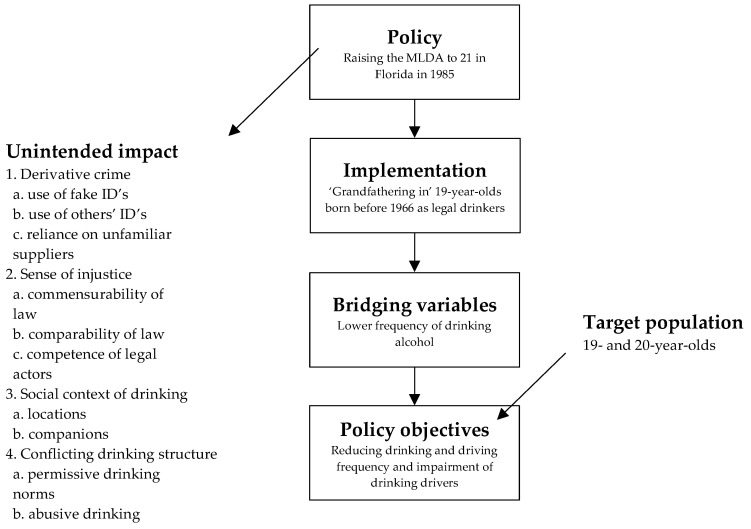
A conceptual model for explaining the impact of raising an minimum legal drinking age (MLDA) [19] (copyright© Academy of Criminal Justice Sciences, reprinted by permission of Informa UK Limited, trading as Taylor & Francis Group, www.tandfonline.com on behalf of Academy of Criminal Justice Sciences).

**Figure 2 ijerph-18-01999-f002:**
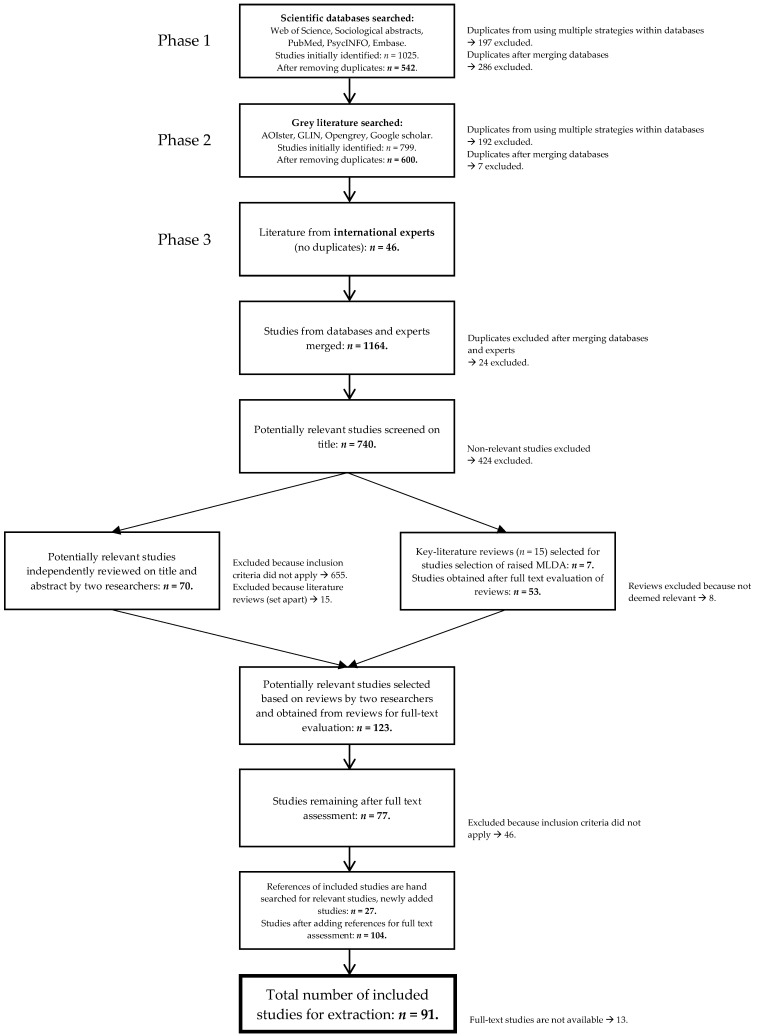
Flow chart of search strategies and results.

**Figure 3 ijerph-18-01999-f003:**
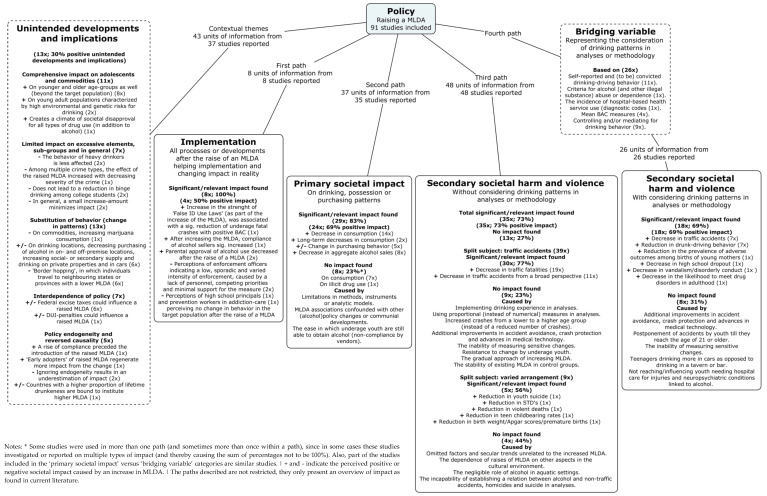
Overview of the impact of raised MLDA as found in current literature.

## Data Availability

Not applicable.

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
