# Peer review of "Examining the Intended and Unintended Impacts of Raising a Minimum Legal Drinking Age on Primary and Secondary Societal Harm and Violence from a Contextual Policy Perspective: A Scoping Review"

_ijerph, 2021, doi:10.3390/ijerph18041999_

Round 1

Reviewer 1 Report

This is a rigorous and comprehensive scoping review that is well presented. The methodological approach is innovative and leads to some useful insights about the topic. The identification of the gap around implementation is particularly useful. I have a few queries and suggestions for the authors which I hope may be useful to improve what is already a strong manuscript.

  1. The unit of analysis in the synthesis is repeatedly referred to as an "insight" but this term is unfamiliar. Could it be defined, perhaps in the Methods? (see point 2 below)
  2. I recommend adding a section 2.5 to the Methods to explain how the synthesis was conducted, expanding on the sentence about thematic content analysis which appears near the end of 2.4. This does not need to be long but would help the reader understand the selection and definition of the 'paths'. Again this term is unfamiliar so some explanation would be helpful. The rationale for division between 'primary and secondary societal impacts' and definition of these could also go here.
  3. The results emphasise vote counting of statistical significance, and the Methods section states that only statistically significant results were extracted for some outcomes. This creates a concern that the findings may reflect and reproduce publication bias. Could this point be addressed in the Discussion?
  4. Related to the point above, the Discussion should describe any limitations of the review. At the same time, it is worth noting that the comprehensive literature search, theory-driven approach and rigorous methods are particular strengths of this review.
  5. regarding page 13 lines 154-155, I am not sure "database analysis" is a research design. The studies cited include recognised research designs such as interrupted time series. I would suggest deleting the sentence as it does not usefully classify the research designs used.
  6. Is Figure 1 the work of the authors or has it been reproduced with permission from reference 19?
  7. regarding page 4 lines 134-135, if the experts were invited then it is not clear how they were also self-selected; please clarify.

Reviewer 2 Report

This study aimed to present a broad spectrum of intended as well as unintended insights derived from the literature relative to the impact of a raise in the minimum legal drinking age on primary and secondary societal harm and violence using the conceptual model proposed by Lanza-Kaduce and Richards. A systematic scoping review was conducted.

Introduction. I suggest to define what is intended and unintended, expected and unexpected impact and consequences of raising a minimum legal drinking age. I.e. is the increase in use of fake ID really an unexpected consequence?

Figures. It is difficult to read and understand text (small font size) in figures (especially Figure 3). Please improve the overall design and quality of figures. I suggest splitting Figure 3 in a few smaller figures.

Results. Please consider shortening the results subsection. It duplicates the content of Figure 3.

Discussion. The limitations and weaknesses of this study are not well described in the discussions. I suggest expanding this section.

Literature. The manuscript deals with an interesting and important topic, and is based on comprehensive literature. I believe the manuscript would add constructively to the existing literature in this field.

Reviewer 3 Report

Thank you for your mail. I've read the manuscript but I haven't introduced any comment yet. My general opinion is that it is a valuable work, interesting and  throughly thought.   -I have missed articles from other countries, such as Australia, which have published tones on alcohol and also on its regulation. -At least to me, it seemed that the theoretical model and methodology are extensively explained, but the explanation was not very concrete and understandable.  I have gone looking for the original framework to understand it better.
  Thank you for the opportunity to review this work. Hope you finish the process successfully  
